# Assessment of Risk Hospitalization due to Acute Respiratory Incidents Related to Ozone Exposure in Silesian Voivodeship (Poland)

**DOI:** 10.3390/ijerph17103591

**Published:** 2020-05-20

**Authors:** Ewa Niewiadomska, Małgorzata Kowalska, Adam Niewiadomski, Michał Skrzypek, Michał A. Kowalski

**Affiliations:** 1Department of Epidemiology and Biostatistics, Faculty of Health Sciences in Bytom, Medical University of Silesia, 40-055 Katowice, Poland; eniewiadomska@sum.edu.pl (E.N.); mskrzypek@sum.edu.pl (M.S.); 2Department of Epidemiology, Faculty of Medical Sciences in Katowice, Medical University of Silesia, 40-055 Katowice, Poland; mkowalska@sum.edu.pl; 3Institute of Information Technology, Lodz University of Technology, 90-924 Łódź, Poland; Adam.Niewiadomski@p.lodz.pl; 4Environmental Exposure Assessment Group, Institute of Epidemiology, Helmholtz Zentrum München, 85764 Neuherberg, Germany

**Keywords:** ozone, health risk, Almon Distributed Lag Model, the Poisson Distributed Lag Model, and Distributed Lag Non-Linear Model

## Abstract

The main aim of this work is the estimation of health risks arising from exposure to ozone or other air pollutants by different statistical models taking into account delayed health effects. This paper presents the risk of hospitalization due to bronchitis and asthma exacerbation in adult inhabitants of Silesian Voivodeship from 1 January 2016 to 31 August 2017. Data were obtained from the daily register of hospitalizations for acute bronchitis (code J20–J21, International Classification of Diseases, Tenth Revision – ICD-10) and asthma (J45–J46) which is governed by the National Health Fund. Meteorological data and data on tropospheric ozone concentrations were obtained from the regional environmental monitoring database of the Provincial Inspector of Environmental Protection in Katowice. The paper includes descriptive and analytical statistical methods used in the estimation of health risk with a delayed effect: Almon Distributed Lag Model, the Poisson Distributed Lag Model, and Distributed Lag Non-Linear Model (DLNM). A significant relationship has only been confirmed by DLNM for bronchitis and a relatively short period (1–3 days) from exposure above the limit value (120 µg/m^3^). The relative risk value was RR = 1.15 (95% CI 1.03–1.28) for a 2-day lag. However, conclusive findings require the continuation of the study over longer observation periods.

## 1. Introduction

Air quality in Poland is being continuously monitored with the State Environmental Monitoring system [1], as well as with networks of unofficial portable sensors to control local air quality levels. Measured values are distributed into the public domain through reports, available on websites and, nowadays, within specially designed software and applications on mobile devices (cellular phones, smartphones) [2]. Those data are especially publicized during worse aero-sanitary conditions and winter smog episodes. Recently, there have been many publications regarding respiratory system condition of inhabitants in large agglomerations located in the southern part of Poland [3]. The upper Silesian urban area is a region with air pollution limit values for PM10 (particulate matter less than 10 μm in diameter) exceeding 100 days in a year [4]. From a public health perspective, this tropospheric ozone concentration has a large impact on the respiratory system [5].

Ozone is often considered as a secondary type of ambient air pollutant, which is mostly related to its precursors’ emission (i.e., nitrogen oxides and volatile organic compounds) in the presence of solar light and high temperature in the summer period [5,6]. Previous studies have demonstrated ozone’s ability to incite an inflammatory response in the respiratory system and consequently an increase in all respiratory, asthma, and acute respiratory infection visits, especially in the warmer season [7,8]. This problem has not been investigated in Poland, nevertheless, this tasks presents an important concern due to repeated adverse climatic conditions, like heat waves and accordingly elevated tropospheric ozone concentrations exceeding the EU information threshold value (1 h concentration 180 µg/m^3^) and maximum daily 8-h mean concentration (120 µg/m^3^) [5,9,10].

Usually, time series are used to assess the impact of short-term human exposure on mortality or morbidity. Various types of mathematical models are used to model dependencies, taking into account the delay of the health effect. The most common are the Almon Distributed Lag Model [11], the Poisson Distributed Lag Model [12], and Distributed Lag Linear and Non-Linear Model (DLNM) [13].

The main aim of this work is estimation of health risk arising from exposure to ozone or other air pollutants by different statistical models taking into account delayed health effect. This paper presents the risk of hospitalization due to bronchitis and asthma exacerbation in adult inhabitants of the Silesian Voivodeship in the years 2016–2017.

## 2. Material and Methods

### 2.1. Health Data

For the purpose of the study, secondary epidemiological data obtained from the daily hospitalization registry for acute bronchitis (J20–J21, ICD-10) and asthma (J45–J46, ICD-10) were taken into account while evaluating the short-term health effect of ozone exposure. Only hospitalizations concerning residents of the central area of the Silesian agglomeration (Silesian Voivodeship, the southern part of Poland) were included in the database. The surveyed agglomeration (the central area of the agglomeration—CAA) includes 14 cities with an area of 3337 km^2^ and a total population of *N* = 1,871,460 in 2016 [14]. A detailed description of the territorial range was presented in a previous publication on exposure to fine particulate matter [3]. Health data from 1 January 2016 to 31 August 2017 were obtained from the National Health Fund database in Katowice. It should be noted that no influenza epidemic was observed during the investigated period. The study is not a medical experiment and did not require the consent of the Bioethics Committee.

### 2.2. Environmental Data

Meteorological data and tropospheric ozone concentrations were obtained from the regional environmental monitoring database of the Provincial Inspector of Environmental Protection in Katowice [15]. The 24-hourly average measurements of O_3_ (μg/m^3^) ozone concentrations measured on an hourly basis and 8-hourly average O_3_ (8 h) concentrations available at 2 automatic stations in the CAA agglomeration, in Katowice (Kossutha Street) and Zabrze (Curie-Skłodowska Street), were taken into account. Due to high correlation coefficients for measurements at both stations [3], values measured at Katowice station were used for further analyses. Among the meteorological data, the following were analyzed: Ambient air temperature and relative humidity, and wind speed, which may favor the formation of photochemical smog. The statistical models also took into account seasonality, with a division of the year into spring, summer, autumn, and winter season. Besides, a classification in terms of climate was used, distinguishing pre-spring, spring, summer, autumn, pre-winter, and winter. Due to the estimation of the risk of acute health events requiring hospital treatment, the models included the division into 7 weekdays, not investigating days off work during which patients are not admitted.

### 2.3. Data Analysis

The descriptive and analytical statistical methods applied to estimate the health risk of people with delayed effects related to environmental exposures are taken into account in this paper. The general characteristics and available statistical methods limited to modeling with delayed effect are presented: the Almon Distributed Lag Model, the Poisson Distributed Lag Model, and Distributed Lag Linear and Non-Linear Model (DLNM). Available tools of the R package v.3.6.2 (2019, The R Foundation for Statistical Computing, GNU General Public License; The Comprehensive R Archive Network), were used [13,16,17,18]. The criterion *α =* 0.05 is taken for statistical inference. The assessment of raw data on the level of pollution and meteorological conditions that cause the so-called photochemical smog is the starting point for the descriptive analysis presented here. The environmental factors are compared with the explained values *y_t_,* (here: the number of health services) in the distinguished periods to underline the delayed effect and to determine the maximum possible lag of the health effect *L*. The results are presented are as scatter charts.

### 2.4. The Applied Models

#### 2.4.1. The Almon Distributed Lag Model

The analysis of the exposure-response relation based on distributed lag models (DLM) is chosen, including the originally developed Almon’s method [11]. In its general form, model (1) is represented by the values *y_t_* at the moment/day *t,* explained by linear combinations of functions of the variable *x_t_:*
$p_{l}(x_{t})$, expressing the lag related to moment *t* by period *l* where *l = 0,…,L, L*—maximum lag with error $\varepsilon_{t}$:(1)$$y_{t}=\varepsilon_{t}+\overset{L}{\underset{l=0}{\sum }}\beta_{l}p_{l}(x_{t}).$$

Coefficients $\beta_{l}$. determining the impact of changes in *x_t_* on the expected value of *y_t_*, using the function (2) allows us to reduce the impact of collinearity of variables at the analyzed moment [11,16].
(2)$$\beta_{l}=\alpha_{0}+\alpha_{1}l+\alpha_{2}l^{2}+\ldots +\alpha_{p}l^{p},\;l=1,2,\ldots ,L;\;p=1,2,\ldots ,n;\;n<l.$$

In this model, it is assumed that the influence of the explanatory variable *x_t_* at the explained variable *y_t_* lasts for at most *L* periods. The main problem, however, is the difficulty in choosing a degree of polynomial *p* and lack of interpretation of the impact of many variables. In case of difficulties in defining the *p* value, the use of Koyck transformation is proposed.

As the criterion of choosing a statistical model and, so, the degree of the polynomial and the maximum lag, the Akaike Information Criterion (AIC) is applied usually.

#### 2.4.2. The Poisson Distributed Lag Model

The model using properties of the Poisson distribution is apied because it is similar to the distribution of the explained variable [17]. In its general form, model (3) is represented by the logarithm of the expected values *y_t_* at the moment/day *t*, explained by the linear combination of the i-th predictors functions $p_{i,l}(x_{t})$ expressed by the lag related to moment *t* by period *l*, where *l = 0,…,L, L*—maximum lag, with error $\varepsilon_{t}$:(3)$$\log\left(E\left(y_{t}\right)\right)=\varepsilon_{t}+\overset{m}{\underset{i=1}{\sum }}\beta_{i,l}p_{i,l}(x_{t}).$$

Due to the overdispersion of the model, if the residual deviation is greater than the number of degrees of freedom, the quasi-Poisson regression model is used. As the criterion of choosing a statistical model, and also the model fit indicator, the counterpart ofhe Akaike Information Criterion (AIC) is used—quasi-AIC.

The advantage of the Poisson regression model is the ability to estimate the relative risk (RR) of the health effect with the increase of exposure by unit *Δx_i_ = const*, while this risk is evaluated via formula (4).
(4)$$RR_{i,l}=e^{const\cdot \beta_{i,l}}.$$

#### 2.4.3. Distributed Lag Non-linear Models—DLNM

Because of the limitations mentioned above, a new method is introduced that uses the *spline functions* (splines) in the analysis of time-series data. In its general form, the model (5) is represented by the logarithm of the expected values *y_t_* at the moment/day *t,* explained by the linear combination of predictors $p_{i}(x_{t})$. at the moment *t* and splines $s_{i}\left(x_{t,l}\right)$ expressing a lag concerning the moment *t* by the period *l,* where *l = 0,…,L, L*—maximum lag, with error $\varepsilon_{t}$:(5)$$\log\left(E\left(y_{t}\right)\right)=\varepsilon_{t}+\overset{m}{\underset{i=1}{\sum }}\beta_{i}p_{i}(x_{t})+\overset{p}{\underset{i=m+1}{\sum }}s_{i}\left(x_{t,l}\right).$$

The method is based on the Poisson model but the linearity is here partially replaced with nonlinear and delayed effects defined via spline functions. The method is originally implemented by Gasparrini A. and Armstrong B in the package R dlnm [13,18,19,20], hence, this is the unique statistical package available for free (GNU GPL license), that enables non-linear Poisson regression with delayed effects in analysis of time series data. Because of the overdispersion of the model, appearing when the residual deviation is greater than the number of degrees of freedom, the quasi-Poisson regression model is used. As the criterion for choosing a statistical model and the model fit indicator, the counterpart of the Akaike Information Criterion (AIC) is used—asi-AIC.

The application of the DLNM model with delayed health effects creates opportunities to investigate the simultaneous impact of many factors throughout the period considered. Apart from 8 h ozone concentrations in the model, the following predictors are taken into account: Seasonality related to climatic seasons of the year, day of the week variability, temperature, relative humidity, and wind speed. Specifying the structure of health effect delays for environmental factors is based on spline functions. B-splines are used for temperature, humidity, and wind speed, and based on 3 interpolation nodes defined by the 10th, 50th, and 90th percentile, or 10th, 75th, and 90th percentile. The choice of functions depends on the quasi-AIC criterion. For the concentration of ozone, due to the values in time, a natural spline is adapted with 3 interpolation nodes given above. As in the previous models, the maximum lag of 21 days is taken. Besides, a natural spline is introduced for seasonality taking into account the division into 12 interpolation nodes resulting from climate changeability of 7 seasons over the year in the examined period of 1.67 years.

## 3. Results

Table 1 presents the variability of ozone concentrations and meteorological parameters in the consecutive summer months and the average number of registered hospitalizations for acute bronchitis (J20–J21) and asthma exacerbations (J45–J46). It should be noted that there were 26 days with 8 h ozone levels above the limit value (120 µg/m^3^) in 2017, thus exceeding the permissible number of 25 days approved by the Ministry of Environment Regulation (OJ 2012 item 1031) [21].

The daily pattern of eight-hour ozone concentrations and accompanying hospitalizations due to asthma and bronchitis during the summer season in 2017 is presented in Figure 1. The detailed observation during this period was made due to the extremely unfavorable aerosanitary situation.

The analysis of data for summer periods in years 2016–2017, conducted using the Almon method with the grade 3 polynomial function, with a maximum delay of 21, showed a slight and insignificant effect of eight-hour O_3_ ozone concentration on the increase in the number of hospitalizations due to bronchitis and asthma (Figure 2). A significant additive correlation was only confirmed for asthma and a delay of more than 20 days. A significantly lower number of hospitalizations for asthma occurred in the first days (from 0 to 2 days) after exposure to higher ozone values. Implement a shorter test period led to the ambiguity of the equations for regression coefficients $\beta_{l}$.

The use of Poisson’s model with a delayed health effect allowed us to study the multifactorial impact on registered hospitalizations throughout the study period (1 January 2016–31 August 2017). In addition to the eight-hour ozone concentration, the model took into consideration the seasonality associated with the astronomical season, the variability resulting from the day of the week and meteorological conditions (temperature, relative humidity, and wind speed). No significant effect of ozone concentration increase on the change in risk of hospitalization due to asthma or bronchitis at first 9 days was found (Figure 3). In the case of bronchitis, the significantly ler health risk is observed after 10 days. It is worth adding that the limitation of the study period only to summer 2017 led to a deterioration of the model quality, extended confidence intervals and, finally, to higher values of the AIC criterion.

The relative risk assessment considering the delayed effect (by DLNM model) and 8 h O_3_ concentration over the whole period (January 2016 to August 2017) reveals a strong impact of the concentration of the pollutant during the first days after exposure (Appendix A). An increase of the risk of hospitalization for the low ozone level can be related to the occurrence of high concentrations of particulate matter less than 10 μm in diameter PM_10_ (R_S_ = −0.42; R_S_—Spearman’s correlation coefficient with O_3_(8 h)), particulate matter less than 2.5 μm in diameter PM_2.5_ (R_S_ = −0.59), sulfur dioxide SO_2_ (R_S_ = −0.61), nitrogen dioxide NO_2_ (R_S_ = −0.33), nitrogen oxides NO_X_ (R_S_ = −0.34), carbon monoxide CO (R_S_ = −0.57) during the winter period.

Also, a significant increase in the risk of hospitalization in cases of acute bronchitis occurred the day after the ozone limit value (120 μg/m^3^) was exceeded by one unit (10 μg/m^3^) and persisted for 3 days. No significant increase in the risk of hospitalization was found for acute asthma (Figure 4). There was an increase in the relative risk of hospitalization due to acute bronchitis in the 3 days following the exceedance of the ozone concentration by further units (Appendix A).

## 4. Discussion

All presented models in this paper took into account the delay of the health effect of exposure to eight-hour tropospheric ozone concentrations and the associated meteorological conditions. The results indicate that the DLNM model provides the most sensitive analysis of the relationship between the increase of exposure by a unit (10 µg/m^3^) and the health effect expressed as a daily hospitalization rate due to bronchitis and asthma exacerbation. A significant relationship was confirmed only for bronchitis and a relatively short period (1–3 days) from exposure above the limit value (120 µg/m^3^). The relative risk value was RR = 1.15 (95% CI 1.03–1.28) for a 2-day delay.

In the case of the classic Almon model (1965), a linear dependence of successive predictive delays is used, which, on the one hand, makes it easier to estimate the number of events, but requires controlling the phenomenon of the autocorrelation of individual delays by using the polynomial function. This operation must be preceded by establishing initial conditions, including the maximum delay *L* (affecting the number of variables in the model) and the degree of polynomial *p* for approximation. The resulting system *L* of grade *p* equations requires a sufficiently large data set. The choice of initial conditions is possible due to the application of the model matching criterion, e.g., AIC. A certain limitation of this method is the possibility to assess the influence of only one environmental factor. Nevertheless, the discussed method of data analysis is used in environmental epidemiology works to assess the relationship between the concentration of air pollutants and mortality or hospitalization [22,23].

An alternative solution is the Poisson model based on the classic GLM (Generalized Linear Model) formula, using lags for explanatory variables. Its use in epidemiological studies is closely related to the shape of the distribution of explanatory variables, such as the number of health services or the number of deaths over time. This model enables multifactorial analysis, which makes it suitable for both environmental epidemiology and infectious diseases [3,24]. The selection of the best model is possible with the quasi-AIC model matching criterion. The disadvantage of the discussed method is the model’s tendency to excessive dispersion, i.e., greater variability in the data than in values given on the statistical model.

The relatively newest DLNM method (2006) uses splined functions to define predictors which simplifies specifications of initial conditions for the analysis of relative risk. The model enables the multi-factor analysis and each of the predictors can be implemented by selecting appropriate functions sensitive to individual initial conditions by the quasi-AIC model fit criterion. The flexibility in defining the model and its easy interpretation encourage to use of this method in estimating health risk in environmental epidemiology. The inventors of the method, Armstrong and Gasparrini, demonstrated the significant influence of delayed effects of low and high temperature ranges at the increase of mortality in London in 1997–2012 [25,26,27]. The method DLNM was also used to assess the delay between exposure to air pollution and the risk of preterm delivery [28,29].

A meta-analysis of the methods used in the analysis of time series for infectious diseases carried out in 2015 on a set of 33 international studies shows the predominant share of Poisson’s and quasi-Poisson’s regression, *n* = 31 (93.9%) [30]. In 28 (84.8%) cases the classical formula of the GLM model was considered, while in 3 (9.1%) the GAM (Generalized Additive Models) functions were used. In two cases (6.1%) mixed models were used. The vast majority of works, *n* = 28 (84.8%), used the effect of delays in risk analysis. In this paper, we underline the applications of the methods presented in evaluation of delayed health effect of air pollution and meteorological conditions. The literature review points at frequent using of Distributed Lag Non-linear Models [25,26,27,28,29,30,31] that replaces the formerly-suggested Poisson regression [13,31].

In conclusion, it should be considered that the DLNM model with a delayed health effect is best used for further analyses of the causal relationship between daily ozone exposure and daily hospital admissions due to acute respiratory incidents. Authors of publications summarizing numerous studies in this area indicate that it is best to study short delays (lag0, lag1, or lag0–1) because these longer delays are less frequently analyzed and it becomes impossible to compare the results obtained. Moreover, the results of the meta-analysis indicate that the increase in ozone concentration by 10 µg/m^3^ is associated with a slight but significant risk of asthma-related hospitalization at RR = 1.009 (95% CI: 1.006–1.011) [8]. In our study, the significant increase in the risk of hospitalization in the case of acute bronchitis was observed for 3-day lag delays. Similarly to our study, stronger relations between ozone concentrations and hospitalization of asthma were obtained for non-linear models and simultaneously for 2-day lag delays [32,33,34]. The previously-cited paper also suggests that the risk of asthma-related emergency room visits and hospital admissions is stronger in the case of the longer lag of exposure (lag ≥ 2 days) than in shorter lag of exposure expressed by lag < 2 days, 1.010 (1.006–1.013) and 1.007 (1.004–1.011), respectively [8]. A positive and statistically significant association was observed between ozone and asthma emergency department visits 2 days later, and such as in our study, the strength of the association was higher in nonlinear models [32]. Another study in California documented that ozone-associated increases in medical visits of asthma and acute respiratory infections were slightly larger in the previous 3 days of ozone exposure in a whole year (however, the increase was higher in the warm season) [7]. An increase in 3-day moving average (lag 0–2) ozone concentration leads to a stronger effect in asthmatics with allergic comorbidities than in asthmatics without comorbidities, the adjusted odds ratio of acute asthma visits were 1.08 (95% CI: 1.02, 1.14) and 1.00 (95% CI: 0.95, 1.05), respectively [33]. In Europe and the United States, the largest effects of mortality were observed with ozone exposure over 3 days, whereas in Canada the strongest effect was observed with the average of lags 0–1 [35]. Moreover, the same researchers concluded that this difference is difficult to explain. On the other hand, the results of the Almon model indicate the significantly lower number of hospitalizations for asthma from 0 to 2 days after exposure to ozone values. As we mentioned, the limitation of this method is assessing the influence of only one environmental factor (ozone). The influence of factors: day of the week variability, temperature, relative humidity, and wind speed, was omitted in this model. The inability to include seasonality in the model forced the authors to limit the data to the summer periods only. If these factors were taken into account in the assumptions of this model, it would lead us to different results. Besides, Bhaskaran et al. in his study draws attention to the apparently protective effect at longer lags [24]. This problem is visible in the presented study using the Poisson model. In the case of bronchitis, the significantly lower health risk is observed after 10 days. Recent epidemiological studies considering larger series or using other statistical approaches such as case-crossover design have confirmed that ozone is indeed associated with acute adverse health effects, expressed by morbidity [6]. Previously published data underline that the association between respiratory effects and ozone appears with a lag-effect [36,37,38] and observed changes in lung function after exposure are most likely associated with an exuberant airway inflammatory response [39]. Finally, an important addition to public health information comes from the natural experiment associated with the 1996 Olympic Games in Atlanta—a decrease in ozone concentration resulted in the lowering of asthma hospital admissions [36].

## 5. Limitations

The study design is an ecological type and, according to its nature, is based on secondary epidemiological data. Therefore, certain biases related to the incorrect recording of the cause of hospitalization cannot be excluded. The basic limitation for an unambiguous conclusion is the relatively short observation period (from 1 January 2016 to 31 August 2017), resulting from the limitations of the previously-acquired database of the National Health Fund for the analysis of winter smog and its impact on health. Moreover, the available data only allow the analysis of the number of medical services provided in the whole CAA agglomeration without the possibility to assess the condition in individual districts or separately for gender and individual age groups. Another disadvantage is the method used to measure exposure to ozone; in the studied region O_3_ was measured only at two measurement stations. Nevertheless, due to high values of correlation coefficients between concentrations of pollutants measured at other stations and close to homogeny character of the region in terms of urbanization and car traffic, data from Katowice station which had complete measurements of 8 h tropospheric ozone in the studied period was rather used. In the analysis of time series, measurements averaged up to 8 h are preferred, which is a good practice due to the relatively large daily air temperature amplitudes in the studied region. Despite the limitations mentioned above, it was found that the results are so interesting that it is worthwhile to present them to make it easier for future researchers to choose the right method for estimating potential health risks. Due to the change of climatic conditions and increasing public demand for reliable information about the risk, it is necessary to continue research using the presented models over longer periods.

## 6. Conclusions

It has been revealed that non-linear models and two-day lag delays gave the strongest and statistically significant response to the relationship between ozone concentrations and hospitalization due to bronchitis. In the case of asthma, no significant relationship was confirmed for the studied period. Clear conclusions require further studies with longer observation time.

## Figures and Tables

**Figure 1 ijerph-17-03591-f001:**
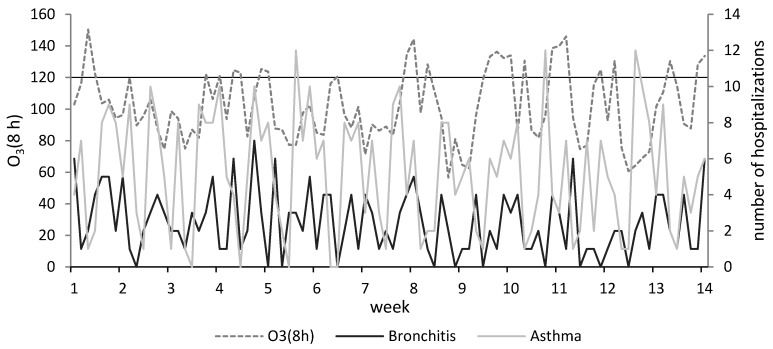
Daily ozone concentrations and the number of hospitalizations due to asthma and bronchitis in summer 2017.

**Figure 2 ijerph-17-03591-f002:**
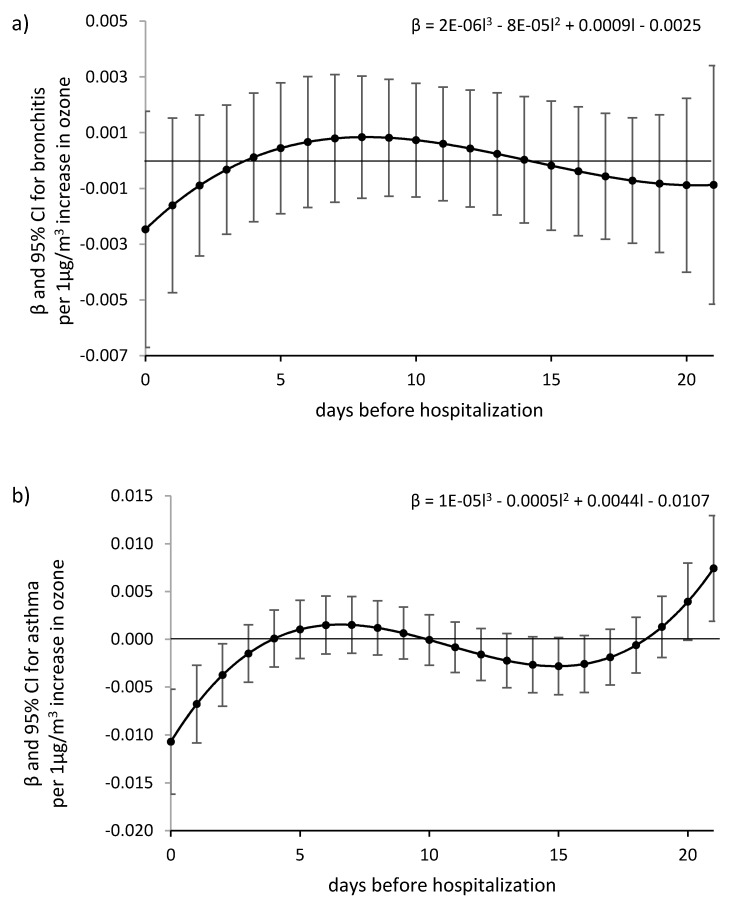
$\beta_{l}$ coefficients for the Almon model with the approximation with 3rd-degree polynomial function, for the number of: Bronchitis (**a**) J20–J21, and asthma (**b**) J45–J46 hospitalization related to increase of 8-h ozone concentration by 1 µg/m^3^. The day 0—the day of registered hospitalization; the day *l*—the day before hospitalization.

**Figure 3 ijerph-17-03591-f003:**
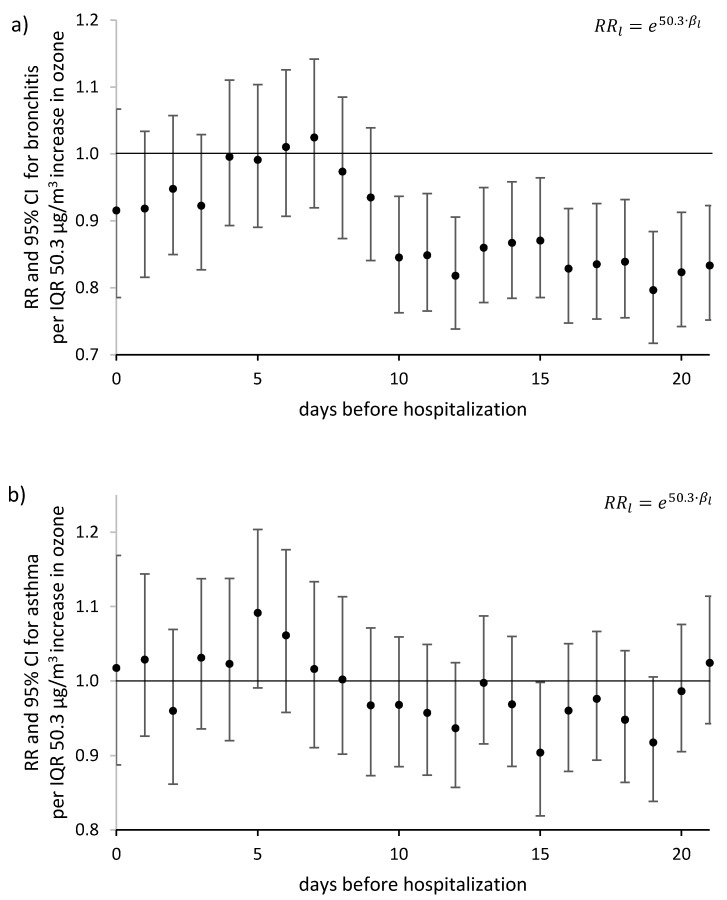
Relative risk $RR_{l}$ of daily hospitalizations due to: (**a**) bronchitis (J20–J21), (**b**) asthma (J45–J46), related to 8 h ozone concentration increase by the interquartile range—IQR 50.3 μg/m^3^. The day 0—the day of registered hospitalization; the day *l*—the day before hospitalization.

**Figure 4 ijerph-17-03591-f004:**
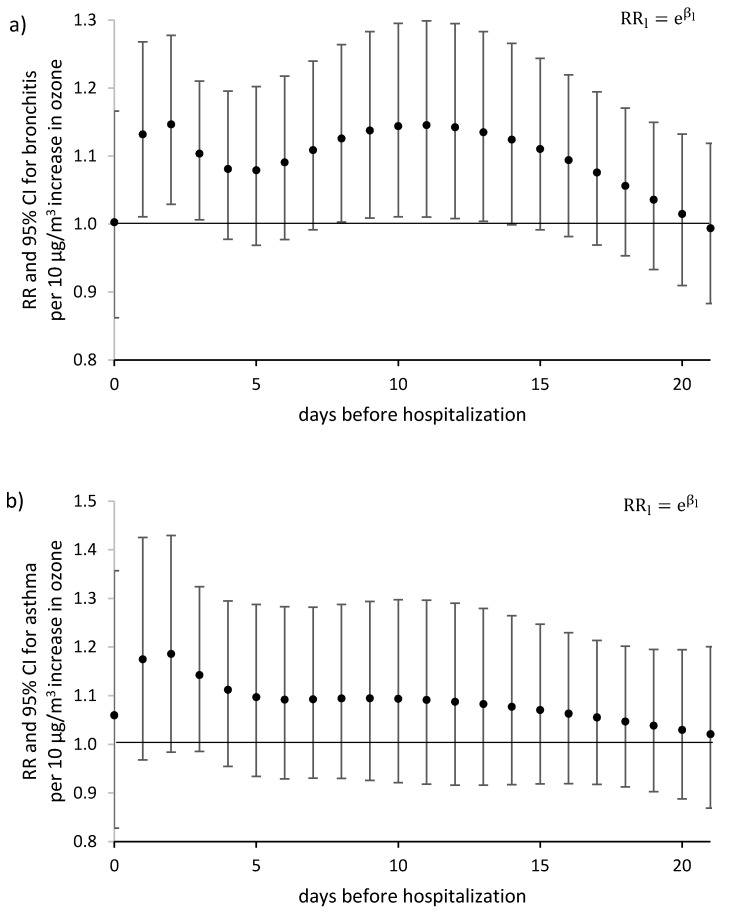
Relative risk $RR_{l}$ of daily hospitalizations due to: (**a**) Bronchitis (J20–J21), (**b**) asthma (J45–J46), associated with an 8 h ozone concentration increase by unit (10 μg/m^3^) compared to a normative value of 120 μg/m^3^. The day 0—the day of registered hospitalization; the day *l*—the day before hospitalization.

**Table 1 ijerph-17-03591-t001:** Meteorological conditions and ozone concentrations, and number of hospitalizations due to asthma and bronchitis in the summer months, the central area of the agglomeration—CAA 2016–2017.

Parameter	Total	2016			2017		
		June	July	August	June	July	August
Air Temperature (°C)	9.2 (13.9) −18.4–26.8	17.5 (4) 14.4–26.2	19.2 (5) 13.6–25	17.4 (4.2) 11.6–23.2	18.4 (4) 14.4–26.2	17.6 (3.9) 11–25.4	19.6 (6.6) 14.4–25.4
Relative Humidity (%)	81.5 (18.6) 48–99	69 (16.8) 55.3–96.5	79.5 (21.5) 60.3–94.5	74.8 (10.3) 66.8–97	67.6 (10.5) 51.3–95	73.7 (13.7) 58–92.7	73.7 (13.7) 60.3–96.3
Wind Speed (m/s)	0.8 (0.8) 0–2.6	0.4 (0.8) 0–1	0.4 (0.8) 0–1.4	0.5 (0.8) 0–1.3	0.8 (0.8) 0–1.8	0.6 (0.6) 0–1.8	0.4 (0.8) 0–1.2
O_3_ (μg/m^3^)	47 (33.3) 3.7–99.3	63.8 (18.3) 36–89.7	57.3 (26) 36–89.7	52 (18) 28–65.7	71.3 (16) 44–99.3	58.7 (15.3) 38.7–98.7	66 (29) 35–94.3
O_3_ 8 h (μg/m^3^)	71.3 (50.3) 4.5–150.3	99.8 (24) 55–149	92 (31.3) 52.7–129.3	83.7 (32) 44–115.7	103.3 (31.3) 74.7–150.3	94.7 (27.7) 56.7–144.3	101.7 (48.7) 60.7–146
Number of Days with O_3_ 8 h ≥ 120 μg/m^3^	44	5	4	0	9	5	12
Number and Percentage of Hospitalizations							
Bronchitis	4674 (100)	96 (2.1)	67 (1.4)	76 (1.6)	88 (1.9)	70 (1.5)	66 (1.4)
Asthma	3815 (100)	153 (4)	153 (4)	165 (4.3)	168 (4.4)	156 (4.1)	154 (4)

The table contains the median (IQR interquartile range) and the min-max range.

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
