# Peer review of "Assessment of Risk Hospitalization due to Acute Respiratory Incidents Related to Ozone Exposure in Silesian Voivodeship (Poland)"

_ijerph, 2020, doi:10.3390/ijerph17103591_

Round 1
Reviewer 1 Report
After reviewing the authors' response, I recommend acceptance.
Author Response
At first, we would like to thank the Reviewer for the detailed analysis of the presented manuscript. We agree with the majority of the comments. Our replies are written in red text. We made several changes in the revised version of the manuscript regarding the Reviewer’s comments, also marked with red text. For a better reading, we added the numbers of pages and the numbers of lines in the document with the new version of the article (Revised Manuscript with Track Changes). We believed that the below explanation will be satisfying for the Reviewer.
Reviewer 2 Report
The authors have been receptive to the comments but I still have the following concerns:
- I find this response concerning: "The authors' intention was to present a negative health impact because of exposure to ozone, hence the conclusions on protection effects were omitted."
I hope the authors mean to say they hypothesized a negative health impact, otherwise this makes it seem like they're trying to hide something. The data are what the data are, if the results are counterintuitive to the initial hypothesis the authors should just do their best to explain them.
- The authors write: “On the other hand, the results of the Almon model indicate the significantly lower number of hospitalizations for asthma from 0 to 2 days after exposure to ozone values. Due to the rare use of this method, these results are difficult to interpret.” This completely undermines why the authors proposed this model in the first place. If it’s rarely used why are you using it? It makes more sense to just compare the underlying assumptions of the 3 models to try to explain why you see different results depending on the model used.
- Again the authors have not adequately labeled their figures. The y axes should say something like RR for bronchitis per x increase in ozone etc
- The DLM and the DLNM are the same thing, this is just a difference in nomenclature. The papers cited used Gasparrini’s DLNM package. I also don’t think it’s necessary to include those citations in the manuscript since they don’t deal with respiratory hospitalizations. The DLNM method has been used frequently enough.
- In figures 3 and 4 include in the legend what method was used to compute the estimates.
Author Response
IJERPH: ijerph-778574
Assessment of risk hospitalization due to acute respiratory incidents related to ozone exposure in Silesian voivodeship (Poland)
At first, we would like to thank the Reviewer for the detailed analysis of the presented manuscript. We agree with the majority of the comments. Our replies are written in red text. We made several changes in the revised version of the manuscript regarding the Reviewer’s comments, also marked with red text. For a better reading, we added the numbers of pages and the numbers of lines in the document with the new version of the article (Revised Manuscript with Track Changes). We believed that the below explanation will be satisfying for the Reviewer.
Re.1. I find this response concerning: "The authors' intention was to present a negative health impact because of exposure to ozone, hence the conclusions on protection effects were omitted."
I hope the authors mean to say they hypothesized a negative health impact, otherwise this makes it seem like they're trying to hide something. The data are what the data are, if the results are counterintuitive to the initial hypothesis the authors should just do their best to explain them.
In our opinion, this remark is a little unfair. We assure you that we didn't want to hide anything. We hypothesized a negative health impact and (as you can see) we presented all observed effects in the Result section and Discussion section:
Page 6, lines: 187-189: “A significant additive correlation was only confirmed for asthma and a delay of more than 20 days. A significantly lower number of hospitalizations for asthma occurred in the first days (from 0 to 2 days) after exposure to higher ozone values.”
Page 6-7, lines: 201-203: “No significant effect of ozone concentration increase on the change in risk of hospitalization due to asthma or bronchitis at first 9 days was found (Fig. 3). In the case of bronchitis, the significantly lower health risk is observed after 10 days.”
Page 10, lines: 301-303: „On the other hand, the results of the Almon model indicate the significantly lower number of hospitalizations for asthma from 0 to 2 days after exposure to ozone values.”
Page 10, lines: 308-311: „Besides, Bhaskaran et al. in his study draw attention to the protective effect at longer lags [24]. This problem is visible in the presented study using the Poisson model. In the case of bronchitis, the significantly lower health risk is observed after 10 days.”
Re. 2. The authors write: “On the other hand, the results of the Almon model indicate the significantly lower number of hospitalizations for asthma from 0 to 2 days after exposure to ozone values. Due to the rare use of this method, these results are difficult to interpret.” This completely undermines why the authors proposed this model in the first place. If it’s rarely used why are you using it? It makes more sense to just compare the underlying assumptions of the 3 models to try to explain why you see different results depending on the model used.
OK, we rearranged somewhat the text, kindly see below.
Page 10, lines: 304-308: “Due to the rare use of this method, these results are difficult to interpret. As we mentioned, the limitation of this method is assessing the influence of only one environmental factor (ozone). The influence of factors: day of the week variability, temperature, relative humidity, and wind speed, was omitted in this model. The inability to include seasonality in the model forced the authors to limit the data to the summer periods only. If these factors were taken into account in the assumptions of this model, it would lead us to different results.”
Re. 3. Again the authors have not adequately labelled their figures. The y axes should say something like RR for bronchitis per x increase in ozone etc.
OK, according to Reviewer expectation the labels of y axes are changed to following:
Page 6, lines: 191-192: “β and 95% CI of bronchitis/asthma hospitalization” and consequently we added the following text “hospitalization related to increasing of 8-hour ozone concentration by 1 µg/m3” in the Figure title (lines 194-196)
Page 7, lines: 206-207: “RR and 95% CI of bronchitis/asthma hospitalization”
Page 8, lines: 225-226: “RR and 95% CI for bronchitis/asthma hospitalization”
Re. 4. The DLM and the DLNM are the same thing, this is just a difference in nomenclature. The papers cited used Gasparrini’s DLNM package. I also don’t think it’s necessary to include those citations in the manuscript since they don’t deal with respiratory hospitalizations. The DLNM method has been used frequently enough.
We think, that we didn't understand the previous remark of the Reviewer. According to current Reviewer suggestion citations [30,31] are excluded. Kindly see the current version of the manuscript body.
Re. 5. In figures 3 and 4 include in the legend what method was used to compute the estimates.
We hope that the following explanation will be accepted by the Reviewer. The legend of the RR computation is presented at the following formulas:
Page 7, lines: 206-207: “”
Page 8, lines: 225-226: “”

This manuscript is a resubmission of an earlier submission. The following is a list of the peer review reports and author responses from that submission.
Round 1
Reviewer 1 Report
Abstract: Need to include all the statistical methods used in the paper, or remove the text "... such as" in the sentence "The paper includes descriptive and analytical statistical methods used in ... such as Almon ..."
Abstract: A statistical relationship was confirmed for bronchitis. However, the model(s) that was used to find this relationship needs to be specified.
Discussion, third paragraph: Replace "thanks" to "with".
Discussion, third paragraph: Further describe the meaning of "excessive dispersion".
Discussion: Describe and rank which models are most common in the peer-reviewed literature for air pollution epidemiological studies with this type of study design.
Author Response
IJERPH: ijerph-778574
Assessment of risk hospitalization due to acute respiratory incidents related to ozone exposure in Silesian voivodeship (Poland)
At first, we would like to thank the Reviewers for the detailed analysis of the presented manuscript. We agree with the majority of the comments. Our replies are written in red text. We made several changes in the revised version of the manuscript regarding the Reviewers’ comments, also marked with red text. For better reading, we added the numbers of pages and the numbers of lines in the document with the new version of the article (Revised Manuscript with Track Changes).
Reviewer #1
Re. 1. Abstract: Need to include all the statistical methods used in the paper, or remove the text "... such as" in the sentence "The paper includes descriptive and analytical statistical methods used in ... such as Almon ..."
Thank you. The text “such as” is removed:
page 1, lines: 27-30: “The paper includes descriptive and analytical statistical methods used in the estimation of health risk with a delayed effect: such as Almon Distributed Lag Model, the Poisson Distributed Lag Model, and Distributed Lag Non-Linear Model (DLNM).”
Re.2. Abstract: A statistical relationship was confirmed for bronchitis. However, the model(s) that was used to find this relationship needs to be specified.
Thank you for the remark. The model is specified:
Page 1, lines: 30-31: “A significant relationship has only been confirmed by DLNM for bronchitis and a relatively short period (1-3 days) from exposure above the limit value (120 µg/m3).”
Re.3. Discussion, third paragraph: Replace "thanks" to "with".
Thank you – we corrected the mistake. The word is changed:
Page 12, lines: 289-291: “The selection of the best model is possible thanks to with the quasi-AIC model matching criterion.”
Re.4. Discussion, third paragraph: Further describe the meaning of "excessive dispersion".
The meaning of “excessive dispersion” is described. Kindly see:
Page 12, lines: 291-293: “The disadvantage of the discussed method is the model tendency to excessive dispersion, i.e. greater variability in the data than in values given on the statistical model.”
We hope that the above explanation will be accepted by the Reviewer.
Re.5. Discussion: Describe and rank which models are most common in the peer-reviewed literature for air pollution epidemiological studies with this type of study design.
We described the most common methods. Kindly see:
Page 12, lines: 303-311: “A meta-analysis of the methods used in the analysis of time series for infectious diseases carried out in 2015 on a set of 33 international studies shows the predominant share of Poisson's and quasi-Poisson's regression, n=31 (93.9%) [32]. In 28 (84.8%) cases the classical formula of the GLM model was considered, while in 3 (9.1%) the GAM (Generalized Additive Models) functions were used. In two cases (6.1%) mixed models were used. The vast majority of works, n=28 (84.8%), used the effect of delays in risk analysis. In this paper, we underline the applications of the methods presented in evaluation of delayed health effect of air pollution and meteorological conditions. The literature review points at frequent using of Distributed Lag Non-linear Models [27-31] that replaces formerly suggested Poisson regression [33].”
References:
Page 16, lines: 475-477: 33. Baxter, L.A., Finch, S.J., Lipfert, F.W., Yu, Q. Comparing estimates of the effects of air pollution on human mortality obtained using different regression methodologies. Risk Anal, 1997, 17(3), 273-8. https://doi.org/10.1111/j.1539-6924.1997.tb00865.x.

Reviewer 2 Report
The authors present an investigation into the association between ozone at different lags and risk of hospitalization for bronchitis and asthma in Poland.
Major comments
I don’t understand the framing of this paper. This is not truly a review because it is not comprehensive about all the methods used to analyze these types of data and it is not comparing new methods to established ones. I don’t have a particular problem with showing the data analyzed in 3 different ways but this looks like the work you do before publishing not what you publish. It would make more sense to write this in the traditional way this is presented, if this type of data have not been analyzed in Poland that is a sufficient impetus for analysis versus comparing already established analytical methods.
Introduction
- “unauthorized measurement sensors” should be replaced by “unofficial” and specify what this means
- Must describe what PM10 is, as well as the other components mentioned in the Results in lines 228 and 229
- Lines 49-50, why are mediterranean countries mentioned here? Poland isn’t one and there is no meaningful connection made.
Methods
- Please remove tables 1-3. This information is unnecessary and detracts from the analyses. It would be sufficient to briefly describe them, their purpose and cite any relevant literature.
Results
- Eliminate division signs on Table 4, these should be dashes.
- Figure 1. Change date formats on bottom, use weeks (week 1, 2, …) is there a particular reason only data from 2017 are presented?
- Fix labels in Figure 2. And 3. (don’t include numbers intertwined with the lines) specify if they are dates (time). Specify time zero is time of event. Add descriptive labels to both x and y axes.
- Lines 212-223, this should be moved to the methods section.
- Lines 224-231, I don’t understand why the data from the complete period is being looked at instead of the summers all of a sudden.
- Remove figure 4 or show the 2-D slice plot for the entire period.
- Figure 5, do the same as for figured 2 and 3
Discussion
- Need to discuss how these results fit in the current literature. Have other papers reported associations with 2-day lags? How do these compare to other European countries or countries with similar ozone levels.
Author Response
IJERPH: ijerph-778574
Assessment of risk hospitalization due to acute respiratory incidents related to ozone exposure in Silesian voivodeship (Poland)
At first, we would like to thank the Reviewers for the detailed analysis of the presented manuscript. We agree with the majority of the comments. Our replies are written in red text. We made several changes in the revised version of the manuscript regarding the Reviewers’ comments, also marked with red text. For better reading, we added the numbers of pages and the numbers of lines in the document with the new version of the article (Revised Manuscript with Track Changes).
Reviewer #2
Re.1. “I don’t understand the framing of this paper. This is not truly a review because it is not comprehensive about all the methods used to analyze these types of data and it is not comparing new methods to established ones. I don’t have a particular problem with showing the data analyzed in 3 different ways but this looks like the work you do before publishing not what you publish. It would make more sense to write this in the traditional way this is presented, if this type of data have not been analyzed in Poland that is a sufficient impetus for analysis versus comparing already established analytical methods.”
Thank you for the comment. Any research on photochemical smog and its health effect have not been run in Poland up to now. Because of increasing heatwaves, we decided to determine the impact of ozone at health of inhabitants of Silesia. Three accessible methods dedicated to this kind of analysis are applied to confirm the hypothesis. We modified somewhat the aim of the study with hope, that our explanation will be accepted by the Reviewer:
Page 1,2, lines: 18-22, 65-69: “The aim of this paper is to review and evaluate the possibility of using available statistical methods to estimate the health risks associated with exposure to ozone and taking into account the delay of the health effect. The main aim of this work is the estimation of health risks arising from exposure to ozone or other air pollutants by different statistical models taking into account delayed health effects.”
Re.2. Introduction: “unauthorized measurement sensors” should be replaced by “unofficial” and specify what this means
The word is changed and the meaning is described. Kindly see:
Page 1, Lines: 39-41: “Air quality in Poland is being continuously monitored with the State Environmental Monitoring system [1], as well as with a networks of unauthorized unofficial portable measurement sensors to control local air quality levels.”
Re.3. Introduction: Must describe what PM10 is, as well as the other components mentioned in the Results in lines 228 and 229
Thank you for the remark. The abbreviations are explained:
Page 2, Lines: 46-47: ”Upper Silesian urban area is a region, with the air pollution limit values for PM10 (particulate matter, diameter 10μm) are exceeded more than for 100 days in a year [4].”
Page 10, Lines: 245-249: “An increase of the risk of hospitalization for the low ozone level can be related to the occurrence of high concentrations of particulate matter, diameter 10μm PM10 (RS=-0.42; RS - Spearman's correlation coefficient with O3(8h)), particulate matter, diameter 2,5μm PM2.5 (RS=-0.59), sulfur dioxide SO2 (RS=-0.61), nitrogen dioxide NO2 (RS=-0.33), nitrogen oxides NOX (RS=-0.34), carbon monoxide CO (RS=-0.57) during the winter period.”
Re.4. Introduction: Lines 49-50, why are Mediterranean countries mentioned here? Poland isn’t one and there is no meaningful connection made.
Thank you for the remark. The sentence is deleted:
Page 2, Lines: 52-53: “Inhabitants of Mediterranean countries often have the highest exposure to ozone [5,6].”
Re.5. Methods: Please remove tables 1-3. This information is unnecessary and detracts from the analyses. It would be sufficient to briefly describe them, their purpose and cite any relevant literature.
We removed Tables 1-3 and their descriptions.
Re.6. Results: Eliminate division signs on Table 4, these should be dashes.
We renamed Table 4 – Table 1 and replaced division signs “÷” by dashes”-“ – page 6, lines: 194-196.
Re.7. Results: Figure 1. Change date formats on bottom, use weeks (week 1, 2, …) is there a particular reason only data from 2017 are presented?
The axis label format was changed – page 7, lines: 201-204. We used labels: week 1, 2, 3 e.t.c.
The data was presented only in 2017 because of high ozone levels in this period. The authors intend to underline especially health effect in that period; it is presented in Table 1 (former Table 4) and explained. Kindly see:
Page 6, lines: 191-193: „It should be noted, that there were 26 days with 8h ozone level above limit value (120 µg/m3) in 2017, thus exceeding the permissible number of 25 days approved by the Ministry of Environment Regulation (OJ 2012 item 1031) [23].”
Besides, the following explanation is added:
Page 7, lines: 199-200: „The detailed observation during this period was made due to the extremely unfavourable aerosanitary situation.”
Re.8. Results: Fix labels in Figure 2. And 3. (don’t include numbers intertwined with the lines) specify if they are dates (time). Specify time zero is time of event. Add descriptive labels to both x and y axes.
The axes labels of the chart 2 and 3 were moved for better visualization (page 8,9). The delay unit in the axis title was added – lag [day]. The “time zero” was specified in the titles of Figure 2 and 3.
Page 8; 9, lines: 214-216; 229-230: “The day 0 is available as the moment of exposure”
Re.9. Results: Lines 212-223, this should be moved to the methods section.
We moved the paragraph to Methods section (Page 6, lines:175-186).
Re.10. Results: Lines 224-231, I don’t understand why the data from the complete period is being looked at instead of the summers all of a sudden.
Thank you for the comment. The matter of choosing a longer period was clarified:
Page 8, lines: 223-225: “It is worth adding that the limitation of the study period only to summer 2017 led to a deterioration of the model quality, extended confidence intervals and, finally, to higher values of the AIC criterion.”
We hope that the above explanation will be accepted by the Reviewer.
Re.11. Results: Remove figure 4 or show the 2-D slice plot for the entire period.
The 3D plots were moved to Appendix 1. 2D slice plots for entire period were presented as Figure 4 (page 10, lines: 252-255). The title of Figure 4 was changed:
Page 10, lines: 254-255: “The total estimation of the daily number of hospitalizations yt±SE due to: a) bronchitis (J20-J21), b) asthma (J45-J46), related to eight-hour O3 ozone concentration.”
Re.12. Results: Figure 5, do the same as for figured 2 and 3
The delay unit in the axis title was added – lag [day] (page 11, lines:262-266). The “time zero” was specified in the title of Figure 5 (page 11, lines:264-266).
Re.13. Discussion: Need to discuss how these results fit in the current literature. Have other papers reported associations with 2-day lags? How do these compare to other European countries or countries with similar ozone levels.
Thank you for the comment. In Discussion section we discussed in detail the problem of delayed health effect. Kindly see:
Pages 12-13, lines: 325-346: Previously cited paper suggest also, that risk of asthma-related emergency room visits and hospital admissions is stronger in case of the longer lag of exposure (lag≥2 days) than in shorter lag of exposure expressed by lag<2 days, 1.010 (1.006-1.013) and 1.007 (1.004-1.011), respectively [8]. A positive and statistically significant association was observed between ozone and asthma emergency department visits 2 days later, and such as in our study, the strength of the association was higher in nonlinear models [34]. Another study in California documented that ozone-associated increases in medical visits of asthma and acute respiratory infections were slightly larger in the previous 3 days of ozone exposure in a whole year (however the increase was higher in the warm season) [7]. Increase in 3-day moving average (lag 0-2) ozone concentration leads to a stronger effect in asthmatics with allergic comorbidities than in asthmatics without comorbidities, the adjusted odds ratio of acute asthma visits were 1.08 (95% CI: 1.02, 1.14) and 1.00 (95% CI: 0.95, 1.05), respectively [35]. In Europe and the United States, the largest effects of mortality were observed with ozone exposure over 3 days, whereas in Canada strongest effect was observed with the average of lags 0–1 [37]. Moreover, the same researchers concluded that this difference is difficult to explain. Recent epidemiological studies considering larger series or using other statistical approaches such as case-crossover design have confirmed that ozone is indeed associated with acute adverse health effects, expressed by morbidity [6]. Previously published data underline, that the association between respiratory effects and ozone appears with a lag-effect [38-40], observed changes in lung function after exposure are most likely associated with an exuberant airway inflammatory response [41]. Finally, an important to public health information comes from the natural experiment associated with the 1996 Olympic Games in Atlanta – a decrease in ozone concentration resulted in the lowering of asthma hospital admissions [38].
References:
Page 16, lines: 486-498:
- Peng, R.D., Samoli, E., Pham, L., Dominici, F., et.al. Acute effects of ambient ozone on mortality in Europe and North America: results from the APHENA study. Air Quality, Atmosphere, & Health, 2013, 6(2), 445-453. http://doi.org/10.1007/s11869-012-0180-9.
- Friedman, M.S., Powell, K.E., Hutwagner, L., et al. Impact of changes in transportation and commuting behaviors during the 1996 summer Olympic Games in Atlanta on air quality and childhood asthma. Journal of the American Medical Association, 2001, 285, 897–905.
- Ostro, B., Lipsett, M., Mann, J., Braxton-Owens, H., White, M. Air pollution and exacerbation of asthma in African-American children in Los Angeles. Epidemiology, 2001,12, 200–208.
- Mortimer, K.M., Neas, L.M., Dockery, D.W., Redline, S., Tager, I.B. The effect of air pollution on inner-city children with asthma. Eur Respir J, 2002, 19, 699–705.
- Khatri, S.B., Holguin, F.C., Ryan, P.B., Mannino, D., Erzurum, S.C., Teague, W.G. Association of Ambient Ozone Exposure with Airway Inflammation and Allergy in Adults with Asthma. J Asthma, 2009, 46(8), 777–785.
We hope that the above explanation will be accepted by the Reviewer.

Round 2
Reviewer 2 Report
The authors have been very responsive to the comments, however I still have the following concerns:
1. PM is defined incorrectly, should be particulate matter less than 2.5 microns in diameter for PM2.5 and particulate matter less than 10 microns in diameter for PM10.
2. I would also remove mention of what statistical packages can run the models and only reference the statistical package that was used by the authors to run these specific analyses.
3. The author need to be more clear that you also ran a DLNM with the entire period and how many lags did this include. For all other models only 21 days are mentioned. I don’t think it was necessary to include the entire period, the model could have been extended simply to make sure the desired time frame was not being estimated at the end of the DLNM time frame (ie extend the model to 30 days or 60 days )
4. All the figures need better defined labels. I don't understand this "The day 0 is available as the moment of exposure" The x axis should be labeled days before hospitalization correct? Also need to explain why you see a protective effect at days 1 and 2 in figure 2. Please explain why the relative risk decreases. In Figure 2 is the Y axis is change in counts?
5. I don’t understand why the authors say there is no significant associations in figure 3. What you are presenting clearly shows protective effects starting at lag 10 (none of the estimates cross 1). Do you mean you don’t see cumulative associations? This is also a problem throughout the manuscript. What the results are and what the authors are describing are not clear.
6. I also don’t really understand what figure 4 is showing. Please see the manuscripts below to see how to properly graphically display DLNM models :
- Martens DS, Cox B, Janssen BG, Clemente DBP, Gasparrini A, Vanpoucke C,Lefebvre W, Roels HA, Plusquin M, Nawrot TS. Prenatal Air Pollution and Newborns'Predisposition to Accelerated Biological Aging. JAMA Pediatr. 2017 Dec 1;171(12):1160-1167.
- Rosa MJ, Hair GM, Just AC, Kloog I, Svensson K, Pizano-Zárate ML, Pantic I,Schnaas L, Tamayo-Ortiz M, Baccarelli AA, Tellez-Rojo MM, Wright RO, Sanders AP. Identifying critical windows of prenatal particulate matter (PM(2.5)) exposureand early childhood blood pressure. Environ Res. 2020 Mar;182:109073. To see how DLM should be properly displayed.